# GlobalMamba: Global Image Serialization for Vision Mamba

## Abstract

Vision mambas have demonstrated strong performance with linear complexity to the number of vision tokens. Their efficiency results from processing image tokens sequentially. However, most existing methods employ patch-based image tokenization and then flatten them into 1D sequences for causal processing, which ignore the intrinsic 2D structural correlations of images. It is also difficult to extract global information by sequential processing of local patches. In this paper, we propose a global image serialization method to transform the image into a sequence of causal tokens, which contain global information of the 2D image. We first convert the image from the spatial domain to the frequency domain using Discrete Cosine Transform (DCT) and then arrange the pixels with corresponding frequency ranges. We further transform each set within the same frequency band back to the spatial domain to obtain a series of images before tokenization. We construct a vision mamba model, GlobalMamba, with a causal input format based on the proposed global image serialization, which can better exploit the causal relations among image sequences. Extensive experiments demonstrate the effectiveness of our GlobalMamba, including image classification on ImageNet-1K, object detection on COCO, and semantic segmentation on ADE20K.

## 1 Introduction

Mamba (Gu & Dao, 2023; Lieber et al., 2024) has garnered significant interest within the deep learning community recently due to its efficiency. Compared to the widely adopted transformer-based architectures, Mamba reduces the computational complexity from $O(n^2)$ to $O(n)$, where $n$ represents the length of the input sequence, based on State Space Models (SSMs) (Gu et al., 2022; 2021a;b; Gupta et al., 2022). Mamba further accelerates the originally sequential computation of state variables through a series of hardware-friendly algorithms (e.g., parallel scanning) to enhance efficiency in practice. Mamba has demonstrated competitive performance and good potential in various areas such as image representation learning (Zhu et al., 2024; Ma et al., 2024), video understanding (Li et al., 2024), and point cloud analysis (Liang et al., 2024).

Recent efforts introduce Mamba to computer vision by converting image data into one-dimensional token sequences to accommodate its input formats (Zhu et al., 2024; Liu et al., 2024; Huang et al., 2024; Yang et al., 2024). Specifically, they first perform patch embedding to transform images into tokens of a certain resolution, and then sequentially flatten these tokens in a systematic row-wise and column-wise fashion, either across the global scope (Liu et al., 2024) or within a local window (Huang et al., 2024). Although this operation can adapt to various visual tasks, the inherent causal order between image tokens is directly disrupted. Neighboring regions within the spatial domain of image data typically encode similar visual information, whereas characteristics in spatially distant regions may exhibit pronounced dissimilarity, which is commonly referred to as the local invariance property of images. Therefore, the straightforward token flattening procedure may result in parts of patches that were spatially proximate being placed at relatively distant positions in the flattened sequence, and vice versa. This fails to provide an appropriate ordering for the image modeling within the Mamba frameworks. Furthermore, each individual image token of these vision mambas typically possesses only local information and fail to capture global features, thus exhibiting certain deficiencies in terms of modeling capabilities.

Figure 1: Comparisons of different Vision Mamba framerowks. Vim and VMamba adopt a flattening strategy similar to (a) and (b), transmuting two-dimensional images into one-dimensional sequences by row or column, while LocalMamba (c) performs the corresponding flattening within a local window. Notably, these sequences lack the inherent causal sequencing of tokens that is characteristic of the causal architecture of Mamba causal architecture. Differently, GlobalMamba (d) constructs a causal token sequence by frequency, while ensuring that tokens acquire global feature information.

To address this, we propose GlobalMamba, a modified vision mamba model with global image serialization, as shown in Figure 3. We first transform the original image from the spatial domain to the frequency domain via the Discrete Cosine Transform (DCT), thereby acquiring the spectral distribution. We segment the frequency spectrum into multiple intervals, ranging from lower to higher frequencies. We then iteratively group the pixels in the frequency domain within the same frequency band by nullifing the amplitude values corresponding to frequencies that fall outside the designated intervals during the segmentation. Subsequently, we project these segmented spectral representations back into the spatial domain via an inverse transform. Each segment is then individually processed through a tokenization process, yielding a collection of tokens that are representative of the various frequency intervals and possess an expansive global visual receptive field. We arrange these tokens into a unidimensional causal sequence in ascending order of frequency, which is then subjected to the Mamba feature extraction process. Our GlobalMamba constructs a causal token sequence in the order of frequency, allowing the model to understand images in a process similar to humans (i.e., grasping the low-frequency information such as contours before augmenting with detailed information). The tokens used in GlobalMamba are intrinsically associated with discrete frequency intervals, facilitating an enhanced global encapsulation of the spectral information of visual data. In addition, the construction of the causal sequence aligns with the frequency principle of neural networks, which tends to prioritize fitting the low-frequency components of the input data, and low-frequency information often plays a more decisive role in visual tasks such as image classification. We conduct extensive experiments on various tasks to evaluate the effectiveness of our model, including image classification on ImageNet-1K Russakovsky et al. (2015), object detection on COCO Lin et al. (2014), and semantic segmentation on ADE20K Zhou et al. (2019). The consistent improvements (e.g. +0.6% over Vim on ImageNet-1K) compared with the adopted baselines demonstrate the superiority of the proposed GlobalMamba.

## 2 RELATED WORK

**Vision Mambas.** Convolutional Neural Networks (CNNs) and Visual Transformers (ViTs) are the two most commonly used backbones for computer vision. Among them, CNNs have served as the common backbone for most visual tasks over an extended period due to their unique local receptive field prior (He et al., 2016; Liu et al., 2022; Szegedy et al., 2015; Simonyan & Zisserman, 2014). ResNet (He et al., 2016), in particular, has become the most widely used convolutional structure by employing an efficient residual structure to prevent vanishing gradient issues. Additionally, ViTs have emerged as the foundational model architecture with their exceptional scale-up capability and adaptability to multi-modal inputs (Dosovitskiy et al., 2020; Liu et al., 2021; Li et al., 2022). Recently, motivated by the success of Mamba (Gu & Dao, 2023) in natural language processing, several efforts have attempted to apply it to visual understanding tasks (Zhu et al., 2024; Liu et al., 2024; Huang et al., 2024; Yang et al., 2024; Hu et al., 2024; Patro & Agneeswaran, 2024). Typically, Vision Mamba (Vim) (Zhu et al., 2024) constitutes the pioneering effort in the adaptation

of the Mamba architecture for applications within the domain of computer vision, wherein image tokens are flattened into a one-dimensional sequence to adapt to the input format. VMamba (Liu et al., 2024) and LocalMamba (Huang et al., 2024) substantially enrich the serialization process of images by employing strategies such as multi-directional scanning and local window scanning to enhance the corresponding feature extraction ability. In addition, ZigMa (Hu et al., 2024) further applies the Mamba architecture to visual generation. Nevertheless, the approaches employed in these studies necessitate the flattening of tokens during image processing, thereby undermining the intrinsic local invariance characteristics inherent within images. Consequently, the resultant one-dimensional token sequences are devoid of the causal relationships that should exist between preceding and succeeding elements, as well as between contiguous tokens. Moreover, these flattened tokens are imbued with spatially confined information, lacking a comprehensive grasp of the global context. Addressing this deficiency and enhancing the preservation of such causalities as well as global perceptions constitutes a principal objective of our proposed method.

**Causal Sequence Modeling.** Recurrent Neural Networks (RNNs) (Jordan, 1997; Hochreiter & Schmidhuber, 1997; Cho, 2014) represent the pioneering architectural paradigm within the deep learning domain that inherently captures sequential causal relationships. They take sequential data as input and perform recursion along the progression of the sequence, with all nodes connected in a chain-like structure. Therefore, RNNs are particularly suitable for natural language and time series data samples, which inherently possess temporal causality. Mamba (Gu & Dao, 2023) possesses intermediate hidden state variables similar to RNNs, and the iterative manner between state variables also follows a temporal sequence. Therefore, it lacks rationality to model visual tokens without causal order using Mamba. Causal sequence modeling also exists in the decoder part of transformers (Kim et al., 2018). Currently, large language models widely adopt a decoder-only architecture, utilizing next-token prediction for feature extraction of causal input sequences (Radford, 2018; Radford et al., 2019; Brown, 2020; Touvron et al., 2023a;b; Dubey et al., 2024), which is applicable to both language understanding and generation tasks. However, the direct application of a decoder-only architecture to visual classification tasks does not yield impressive results, with a decrease in accuracy compared to its counterpart with global attention interactions (Chen et al., 2020). In addition, Tian *et al.* (Tian et al., 2024) transformed the original next-token prediction into next-scale prediction to enhance the causality between sequences, thereby improving the quality in visual generation. In this paper, we reinforce the causality between image sequences by frequency segmentation, enhancing their compatibility with subsequent modeling procedures

**Frequency Analysis.** Frequency analysis exhibits profound potential for advancement within the domain of deep learning and computer vision. A collection of scholarly endeavors has delved into the frequency principle, suggesting that neural networks exhibit a learning bias towards preferentially fitting the low-frequency signals in the data (Xu et al., 2019; 2024; Luo et al., 2019). Concurrently, they employ the frequency principle to execute sophisticated interpretive analyses of deep learning and guide the corresponding training process. Additionally, several works leverage frequency analysis to facilitate the practical application of visual tasks (Liang et al., 2023; Xu et al., 2020; Qin et al., 2021; Rao et al., 2023; Xie et al., 2021). For example, Xu *et al.* (Xu et al., 2020) discovered that CNNs exhibit a heightened sensitivity to low-frequency channels and mitigate the loss of information due to spatial downsampling by employing feature selection strategies within the frequency domain. Rao *et al.* (Rao et al., 2023) constructed a GFNet capable of modeling long-term spatial dependencies in the frequency domain with log-linear complexity. In this paper, we utilize the division of frequencies to construct visual token sequences, such that the modeling of Mamba adheres to a causal order from low to high frequencies, which alleviates the destruction of image local invariance to a certain extent. Each image token can also focus more intently on the global information within its corresponding frequency band, offering a superior alternative to previous vision models where tokens only encapsulate local information.

## 3 PROPOSED APPROACH

In this section, we first provide a brief introduction to the preliminaries of Mamba. Subsequently, we elaborate on the specific principle and operational process of frequency segmentation. Lastly, we present an overview of GlobalMamba accompanied by corresponding analysis.

Figure 2: The frequency-based global tokenization of GlobalMamba involves frequency-segmenting images into multiple bands, downsampled and tokenized with a lightweight CNN into casual sequences for subsequent processing.

## 3.1 PRELIMINARIES

**State Space Models.** State space models (SSM) employ intermediate hidden states in accordance with the input sequences, with each state transition being derived from the current input and the output at each time step is jointly determined by the current input and the hidden state:

$$h^{'}(t) = \mathbf{A}h(t) + \mathbf{B}x(t), \qquad y(t) = \mathbf{C}h(t) + \mathbf{D}x(t), \tag{1}$$

where $x(t) \in \mathbb{R}, y(t) \in \mathbb{R}, h(t) \in \mathbb{R}^N$ denote the input, output, and hidden state, respectively. $\mathbf{A} \in \mathbb{R}^{N \times N}, \mathbf{B} \in \mathbb{R}^{N \times 1}, \mathbf{C} \in \mathbb{R}^{1 \times N}, \mathbf{D} \in \mathbb{R}^{1 \times 1}$ are the corresponding learnable parameters to determine the evolution and projection processes. Note that $\mathbf{D}$ is often ignored for brevity.

To apply the aforementioned model to actual discrete data, the zero-order hold technique is employed to discretize the equations. ($\mathbf{A}$ and ($\mathbf{B}$) are transformed into their corresponding discrete forms $\overline{\mathbf{A}}$ and $\overline{\mathbf{B}}$ with a time-scale parameter $\mathbf{\Delta} \in \mathbb{R} > 0$, formulated as follows:

$$\overline{\mathbf{A}} = e^{\mathbf{\Delta A}}, \qquad \overline{\mathbf{B}} = (\mathbf{\Delta A})^{-1}(e^{\mathbf{\Delta A}} - \mathbf{I}) \cdot \mathbf{\Delta B}. \tag{2}$$

The state-space equations with the aforementioned discretization are as follows:

$$h_t = \overline{\mathbf{A}}h_{t-1} + \overline{\mathbf{B}}x_t, \qquad y_t = \mathbf{C}h_t + \mathbf{D}x_t. \tag{3}$$

State space models can be reformulated into a convolutional architecture enabling an efficient training procedure with the time-invariance of the learnable parameters. The specific equation form will not be elaborated for brevity.

**Mamba.** Although time-invariant parameters are beneficial for the efficiency of the training process, he lack of specific differentiation for inputs at varying temporal instances constrains the capacity of the model for feature representation. Consequently, Mamba refines this approach by transitioning from time-invariant to time-variant parameters, modifying the learnable parameters to be relevant to the input data. Specifically, $\mathbf{B}$ and $\mathbf{C}$ are obtained from the input through different linear transformation matrices, while $\mathbf{\Delta}$ is determined by the input undergoing a linear transformation followed by the corresponding activation function. However, the time-variant parameters inherently preclude the model from being transformed into a convolutional form for parallel training. To address this, Mamba employs various hardware optimization algorithms to achieve acceleration, including parallel scanning. Therefore, Mamba maintains training efficiency while constraining the time complexity to $O(n)$, presenting a comparative advantage over the $O(n^2)$ complexity of transformers.

## 3.2 FREQUENCY-BASED GLOBAL IMAGE SERIALIZATION

As mentioned before, Mamba inherently models the input data with a causal order, whereas image tokens, after being flattened by rows and columns, lack the causal relationship between adjacent elements and are therefore not fully applicable to the Mamba architecture. In addition, the tokenization

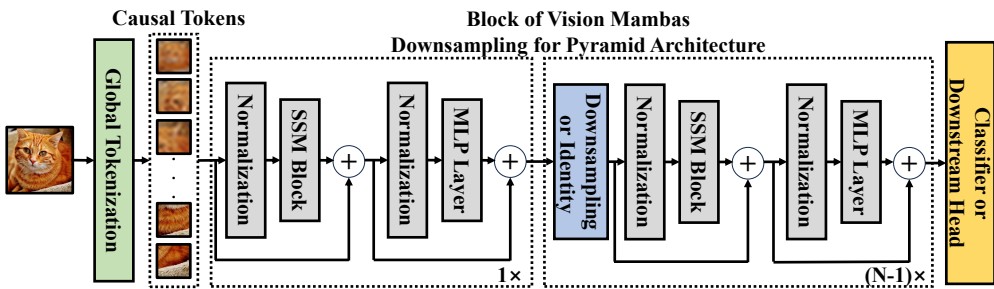

Figure 3: The overall framework of GlobalMamba. The causal sequences obtained through global tokenization will undergo iterative feature extraction via multiple Vision Mamba blocks. Each of these blocks is meticulously designed to incorporate layers of normalization, SSM, and MLP. Feature downsampling might be adopted for pyramid architectures such as VMamba.

process results in separate image tokens representing local features without a more global perception. To address this, we approach the frequency-based global tokenization illustrated in Figure. Given an image $\mathbf{x} \in \mathbb{R}^{h \times w}$, we first utilize the Discrete Cosine Transform (DCT) to convert it into the corresponding frequency domain, with the following formula:

$$F(u,v) = \alpha(u)\alpha(v)\sum_{i=0}^{h-1}\sum_{j=0}^{w-1}\mathbf{x}(i,j)cos(\frac{(2i+1)u\pi}{2h})cos(\frac{(2j+1)u\pi}{2w}), \qquad (4)$$

where $\mathbf{x}(i,j)$ denotes the pixel value at position $(i,j)$. $u$ and $v$ represent the frequency variables, with their ranges from 0 to $h-1$ and from 0 to $w-1$, respectively. $F(u,v)$ is the coefficients after the two-dimensional DCT transformation. $\alpha(u)$ and $\alpha(v)$ are scaling factors, defined as:

$$\alpha(u) = \frac{1}{\sqrt{h}} \quad \text{when } u = 0, \quad \alpha(u) = \frac{1}{\sqrt{2h}} \quad \text{otherwise.} \qquad (5)$$

The spectrum diagram after DCT manifests a pronounced clustering of low-frequency coefficients in the upper left quadrant, while high-frequency components scattered in the lower right corner. Concurrently, the spectrum diagram exhibits symmetry with respect to the main diagonal. Considering the hierarchical organization of frequency components, we delineate the spectral map into discrete frequency segments by adhering to a progression from lower to higher frequencies, and in alignment with a orientation orthogonal to the principal diagonal, as illustrated in Figure 2. Specifically, let $K$ denote the number of frequency segments into which the division is to be made. We unecenly partition the principal diagonal into $K$ segments considering that the non-uniformity of frequency distribution, such that the distance from the $k$th division point to the top-left corner is $\frac{1}{2^{K-k}}$ of the entire length of diagonal. Along each of these division points, a perpendicular line is delineated with respect to the principal diagonal. The interval between consecutive perpendicular lines thus defines the spectral domain for each frequency segment, encapsulating the respective frequency distribution within that segment. We denote the maximum frequency corresponding to each division point as $f_k$ and the segmented frequency bands can be represented as $(0, f_1, ..., f_K)$.

Subsequently, we expand these $K$ frequency bands into $K$ corresponding independent spectrum diagrams, denoted as $(F_1(u,v), ..., F_K(u,v))$. Within the $k$th spectral diagram, we retain the frequency values from the direct current component up to the threshold of $f_k$, while resetting the amplitude of larger frequencies to 0. This segmentation approach ensures that the spectral representation maintains discernible semantic integrity upon its inverse transformation back into the spatial domain. Additionally, it adheres to the frequency principle by increasing the proportion of low-frequency components, which is formulated as follows:

$$F_k(u,v) = I(f_k - f(u,v))F_k(u,v), \quad I(x) = 1 \text{ when } x \geq 0 \text{ and } I(x) = 0 \text{ otherwise,} \qquad (6)$$

where $f(u,v)$ represents the frequency at position $(u,v)$.

Ultimately, we project the derived spectral representations from the frequency domain to the original spatial domain through Inverse Discrete Cosine Transform (IDCT), resulting in $K$ images cor-

Table 1: Architectural overview of the GlobalMamba series.

| Layer name | Output size | GlobalMamba-M* | GlobalMamba-T* | Output size | GlobalMamba-T | GlobalMamba-S | GlobalMamba-B |
|---|---|---|---|---|---|---|---|
| Stem | 14×14 | conv 16×16 dim=192 | conv 16×16 dim=384 | 112×112 | conv 3×3 dim=96 | conv 3×3 dim=96 | conv 3×3 dim=128 |
| Stage 1 | 14×14 | SSM block × 6 dim=192 Identity dim=192 | SSM block × 6 dim=384 Identity dim=384 | 56×56 | SSM block × 2 dim=96 Downsampling dim=192 | SSM block × 2 dim=96 Downsampling dim=192 | SSM block × 2 dim=128 Downsampling dim=256 |
| Stage 2 | 14×14 | SSM block × 6 dim=192 Identity dim=192 | SSM block × 6 dim=384 Identity dim=384 | 28×28 | SSM block × 2 dim=192 Downsampling dim=384 | SSM block × 2 dim=192 Downsampling dim=384 | SSM block × 2 dim=256 Downsampling dim=512 |
| Stage 3 | 14×14 | SSM block × 6 dim=192 Identity dim=192 | SSM block × 6 dim=384 Identity dim=384 | 14×14 | SSM block × 8 dim=384 Downsampling dim=768 | SSM block × 15 dim=384 Downsampling dim=768 | SSM block × 15 dim=512 Downsampling dim=1024 |
| Stage 4 | 14×14 | SSM block × 6 dim=192 | SSM block × 6 dim=384 | 7×7 | SSM block × 2 dim=768 | SSM block × 2 dim=768 | SSM block × 2 dim=1024 |
| Classifier | 1×1 | cls token softmax | cls token softmax | 1×1 | pooling softmax | pooling softmax | pooling softmax |
| Params (M). | | 7 | 26 | Params (M). | 30 | 50 | 89 |
| FLOPs (G) | | 1.7 | 5.7 | FLOPs (G) | 5.3 | 9.5 | 17.0 |

responding to different frequency ranges, presented as follows:

$$\mathbf{x}_k(i,j) = \alpha(u)\alpha(v) \sum_{u=0}^{h-1} \sum_{v=0}^{w-1} F_k(u,v)cos(\frac{(2i+1)u\pi}{2h})cos(\frac{(2j+1)u\pi}{2w}), \qquad (7)$$

where the interpretation of each term is consistent with that in equation 4. With these images corresponding to different frequency bands, we perform spatial downsampling based on the frequency range of each sample, representing images with a smaller frequency range using a lower spatial resolution, formulated as follows:

$$\mathbf{x}'_k = G(\mathbf{x}_k, \frac{h}{2^{K-k}}, \frac{w}{2^{K-k}}), \qquad (8)$$

where $G(\cdot)$ denotes the downsampling interpolation function and $(\frac{h}{2^{K-k}}, \frac{w}{2^{K-k}})$ is the corresponding spatial resolution after downsampling. Subsequently, we proceed with the tokenization procedure, employing an identical compact CNN and a linear module to segment the image samples into patches. These extracted tokens are sequentially organized in a causally ordered manner, progressing from the lower to the higher frequency spectrum.

### 3.3 GLOBALMAMBA

The serialized image tokens are optimally conducive for feature extraction via the sophisticated vision mambas, in which SSM-based encoders are stacked iteratively to extract image features, as shown in Figure 3. The general representation computation can be formulated as follows:

$$\begin{aligned} \mathbf{t_n} &= \mathbf{z}_{n-1} + SSM(Norm(\mathbf{z}_{n-1})), \\ \mathbf{z_n} &= \mathbf{t}_n + MLP(Norm(\mathbf{t}_{n-1})), \end{aligned} \qquad (9)$$

where $\mathbf{z}_n$ denotes the output feature of the $n$th block. Note that vision mambas encompass two distinct architectural paradigms: the pyramid and the plain types. Specifically, the pyramid architecture is characterized by the periodic application of downsampling operations on feature maps between consecutive blocks, whereas the plain architecture is composed exclusively of Identity mappings and the corresponding feature aggregation requires the class token concatenation with the input sequences. Ultimately, the output from the final block will be employed on image classification or

other downstream tasks. Our proposed GlobalMamba is applicable to diverse vision mamba architectures and we present different specifications of GlobalMamba in Table 1.

**Analysis.** Indeed, while the GlobalMamba framework we propose may theoretically result in an expansion of the token sequence, in practical application, this impact is negligible. This is attributable to the application of a significantly higher downsampling factor $2^{K-k}$ to the images within the lower frequency spectrum, which, subsequent to the patchification process, leads to a substantial reduction in the number of generated tokens. This strategic approach effectively curtails the overall length of the sequence, maintaining an optimized balance between computational efficiency and representational integrity. For instance, when $K = 4$ and a standard $16 \times 16$ tokenization procedure is employed, the sequence length yielded by the GlobalMamba approach is marginally higher than that of the conventional baseline method by approximately 30%. Furthermore, our experiments have also demonstrated that simply replicating and expanding the sequence length of baselines does not confer a performance improvement, thereby validating the efficacy of GlobalMamba.

In addition, the tokens procured via GlobalMamba inherently encode more global information, particularly within the low-frequency spectral segments. In instances where $k \geq 4$, the resultant tokens are singular in number, endowing this single token with the ability to represent the global spatial features of that frequency band. At the same time, our method follows a causal order of frequencies from low to high and explicitly increases the proportion of low-frequency information in the entire token sequence. This approach is consistent with the human visual comprehension process and the frequency prior principle of neural networks, which tend to prioritize the learning of low-frequency features to secure a comprehensive understanding before fitting the high-frequency parts for detailed information. Significantly, the low-frequency component often exerts a predominant influence on the interpretive capabilities required for task comprehension.

## 4 EXPERIMENTS

In this section, we conducted extensive experiments to demonstrate the effectiveness of GlobalMamba. We initially trained on ImageNet-1K for image classification and then transferred the pre-trained model to downstream tasks such as object detection and semantic segmentation. Additionally, we provided a series of ablation studies for comparative analysis and investigation. All our experiments were conducted on 8 RTX 3090 GPUs.

### 4.1 IMAGE CLASSIFICATION

We assessed the performance of GlobalMamba on classification tasks using the ImageNet-1K (Russakovsky et al., 2015) dataset, which encompasses over 1,280,000 training samples spanning 1,000 categories, while the validation set comprises 50,000 images. We adopted Vision Mamba (Vim) (Zhu et al., 2024) and VMamba (Liu et al., 2024) as our baselines, maintaining consistent settings for data augmentation and optimizer choices. We categorized the models based on the size of their parameters into GlobalMamba-M (Mini), GlobalMamba-T (Tiny), GlobalMamba-S (Small), and GlobalMamba-B (Base), presented in Table 1. We set the number of training epochs to 300 and employed a cosine schedule for learning rate adjustment. We compared methods with similar parameters and provided both Top-1 accuracy and FLOPs metrics. The experimental results are presented in Table 2, in which the GlobalMamba models marked with * represent the plain structure applied to Vim, while the others represent the pyramid structure applied to VMamba. We observe that GlobalMamba consistently achieves improved accuracy compared to the baseline methods. For instance, on the VMamba-S and VMamba-B models, our method increases the classification accuracy by 0.3% and 0.2%, respectively, thus demonstrating the effectiveness of the proposed GlobalMamba approach. In addition, GlobalMamba entails a marginal increment in FLOPs with the slight expansion of the token sequence length, as analyzed in Section 3.3.

### 4.2 OBJECT DETECTION

We conducted evaluations on MSCOCO2017 (Lin et al., 2014) for object detection and instance segmentation, which comprises over 118,000 training images, 5,000 validation images, and more than 40,000 test images. We employed Mask-RCNN as the detector and performed both 1x and 3x training schedules with the MMDetection (Chen et al., 2019) codebase. We reported the comparison

Table 2: Classification results (%) on ImageNet. († denotes our reproduced performances.)

| Method | Backbone | Image Size | Params (M). | FLOPs (G) | Top-1 Acc |
|---|---|---|---|---|---|
| ResNet-18 (He et al., 2016) | ConvNet | $224^2$ | 12 | - | 69.8 |
| DeiT-T (Touvron et al., 2021) | Transformer | $224^2$ | 6 | 1.3 | 72.2 |
| PlainMamba-L1 (Yang et al., 2024) | SSM | $224^2$ | 7 | 3.0 | 77.9 |
| EffVMamba-T (Pei et al., 2024) | SSM | $224^2$ | 6 | 0.8 | 76.5 |
| EffVMamba-S (Pei et al., 2024) | SSM | $224^2$ | 11 | 1.3 | 78.7 |
| LocalVim-T (Huang et al., 2024) | SSM | $224^2$ | 8 | 1.5 | 76.2 |
| Vim-T† (Zhu et al., 2024) | SSM | $224^2$ | 7 | 1.5 | 75.8 |
| **GlobalMamba-M* (ours)** | SSM | $224^2$ | 7 | 1.7 | **76.4** |
| ResNet-50 (He et al., 2016) | ConvNet | $224^2$ | 25 | - | 77.2 |
| RegNetY-4G (Radosavovic et al., 2020) | ConvNet | $224^2$ | 21 | 4.0 | 80.0 |
| DeiT-S (Touvron et al., 2021) | Transformer | $224^2$ | 22 | 4.6 | 79.9 |
| Swin-T (Liu et al., 2021) | Transformer | $224^2$ | 29 | 4.5 | 81.2 |
| PlainMamba-L2 (Yang et al., 2024) | SSM | $224^2$ | 25 | 8.1 | 81.6 |
| EffVMamba-B (Pei et al., 2024) | SSM | $224^2$ | 33 | 4.0 | 81.8 |
| LocalVim-S (Huang et al., 2024) | SSM | $224^2$ | 28 | 4.8 | 81.2 |
| Vim-S† (Zhu et al., 2024) | SSM | $224^2$ | 26 | 5.1 | 80.3 |
| **GlobalMamba-T* (ours)** | SSM | $224^2$ | 26 | 5.7 | **80.8** |
| VMamba-T (Liu et al., 2024) | SSM | $224^2$ | 30 | 4.9 | 82.6 |
| **GlobalMamba-T (ours)** | SSM | $224^2$ | 30 | 5.3 | **82.8** |
| ResNet-101 (He et al., 2016) | ConvNet | $224^2$ | 45 | - | 78.3 |
| ResNet-152 (He et al., 2016) | ConvNet | $224^2$ | 60 | - | 78.6 |
| RegNetY-8G (Radosavovic et al., 2020) | ConvNet | $224^2$ | 39 | 8.0 | 81.7 |
| Swin-S (Liu et al., 2021) | Transformer | $224^2$ | 50 | 8.7 | 83.2 |
| PlainMamba-L3 (Yang et al., 2024) | SSM | $224^2$ | 50 | 14.4 | 82.3 |
| VMamba-S (Liu et al., 2024) | SSM | $224^2$ | 50 | 8.7 | 83.6 |
| **GlobalMamba-S (ours)** | SSM | $224^2$ | 50 | 9.5 | **83.9** |
| RegNetY-16G (Radosavovic et al., 2020) | ConvNet | $224^2$ | 84 | 16.0 | 82.9 |
| ViT-B/16 (Dosovitskiy et al., 2020) | Transformer | $384^2$ | 86 | 55.4 | 77.9 |
| DeiT-B (Touvron et al., 2021) | Transformer | $224^2$ | 86 | 17.5 | 81.8 |
| Swin-B (Liu et al., 2021) | Transformer | $224^2$ | 88 | 15.4 | 83.5 |
| VMamba-B (Liu et al., 2024) | SSM | $224^2$ | 89 | 15.4 | 83.9 |
| **GlobalMamba-B (ours)** | SSM | $224^2$ | 89 | 17.0 | **84.1** |

results in Table 3. We observe that the SSM-based methods outperform vision transformers under similar parameters, and GlobalMamba consistently achieves better results than VMamba across different model sizes and training settings. For instance, GlobalMamba-S outperforms VMamba-S by 0.3 and 0.2 in box AP under the 1x and 3x schedules, and by 0.2 and 0.1 in mask AP, respectively.

## 4.3 SEMANTIC SEGMENTATION

We adopted ADE20K (Zhou et al., 2019) to verify the effectiveness of GlobalMamba on semantic segmentation. The dataset encompasses 20,210 training images, 2,000 validation images, and 3,000 test images, which are annotated with 150 different semantic categories. We conducted experiments using UPerNet (Xiao et al., 2018) as the segmentor within the MMSegmentation (Contributors, 2020) framework. We employed a training schedule of 160k for comparison, illustrated in Table 4. We find that GlobalMamba achieves certain advantages in terms of both mIoU (SS) and mIoU (MS) compared to VMamba. For example, GlobalMamba-S surpasses the VMamba-S baseline by 0.3 mIoU (SS), which proves the superiority of our proposed framework.

Table 3: Object detection and instance segmentation results on COCO.

| Method | Detector | Params (M). | $AP^b$ | $AP^b_{50}$ | $AP^b_{75}$ | $AP^m$ | $AP^m_{50}$ | $AP^m_{75}$ |
|---|---|---|---|---|---|---|---|---|
| ResNet-50 (He et al., 2016) | MaskRCNN@1x | 44 | 38.2 | 58.8 | 41.4 | 34.7 | 55.7 | 37.2 |
| ResNet-101 (He et al., 2016) | MaskRCNN@1x | 63 | 38.2 | 58.8 | 41.4 | 34.7 | 55.7 | 37.2 |
| ConvNeXt-T (Liu et al., 2022) | MaskRCNN@1x | 48 | 44.2 | 66.6 | 48.3 | 40.1 | 63.3 | 42.8 |
| ConvNeXt-S (Liu et al., 2022) | MaskRCNN@1x | 70 | 45.4 | 67.9 | 50.0 | 41.8 | 65.2 | 45.1 |
| ConvNeXt-T (Liu et al., 2022) | MaskRCNN@3x | 48 | 46.2 | 67.9 | 50.8 | 41.7 | 65.0 | 44.9 |
| ConvNeXt-S (Liu et al., 2022) | MaskRCNN@3x | 70 | 47.9 | 70.0 | 52.7 | 42.9 | 66.9 | 46.2 |
| Swin-T (Liu et al., 2021) | MaskRCNN@1x | 48 | 42.7 | 65.2 | 46.8 | 39.3 | 62.2 | 42.2 |
| Swin-S (Liu et al., 2021) | MaskRCNN@1x | 69 | 44.8 | 66.6 | 48.9 | 40.9 | 63.2 | 44.2 |
| Swin-T (Liu et al., 2021) | MaskRCNN@3x | 48 | 46.0 | 68.1 | 50.3 | 41.6 | 65.1 | 44.9 |
| Swin-S (Liu et al., 2021) | MaskRCNN@3x | 69 | 48.2 | 69.8 | 52.8 | 43.2 | 67.0 | 46.1 |
| VMamba-T (Liu et al., 2024) | MaskRCNN@1x | 50 | 47.3 | 69.3 | 52.0 | 42.7 | 66.4 | 45.9 |
| **GlobalMamba-T (ours)** | MaskRCNN@1x | 50 | **47.6** | **69.4** | **52.2** | **42.9** | **66.5** | **46.0** |
| VMamba-S (Liu et al., 2024) | MaskRCNN@1x | 70 | 48.7 | 70.0 | 53.4 | 43.7 | 67.3 | 47.0 |
| **GlobalMamba-S (ours)** | MaskRCNN@1x | 70 | **49.0** | **70.5** | **53.5** | **43.9** | **67.5** | **47.0** |
| VMamba-B (Liu et al., 2024) | MaskRCNN@1x | 108 | 49.2 | 71.4 | 54.0 | 44.1 | 68.3 | 47.7 |
| **GlobalMamba-B (ours)** | MaskRCNN@1x | 108 | **49.3** | **71.4** | **54.2** | **44.2** | **68.4** | **47.7** |
| VMamba-T (Liu et al., 2024) | MaskRCNN@3x | 50 | 48.8 | 70.4 | 53.5 | 43.7 | 67.4 | 47.0 |
| **GlobalMamba-T (ours)** | MaskRCNN@3x | 50 | **49.0** | **70.5** | **53.7** | **43.8** | **67.5** | **47.1** |
| VMamba-S (Liu et al., 2024) | MaskRCNN@3x | 70 | 49.9 | 70.9 | 54.7 | 44.2 | 68.2 | 47.7 |
| **GlobalMamba-S (ours)** | MaskRCNN@3x | 70 | **50.1** | **80.1** | **54.9** | **44.3** | **68.4** | **47.8** |

Table 4: Semantic segmentation results on ADE20K.

| Method | Segmentor | Image size | Params (M). | mIoU (SS) | mIoU (MS) |
|---|---|---|---|---|---|
| Swin-T (Liu et al., 2021) | UperNet@160k | $512^2$ | 60 | 44.4 | 45.8 |
| Swin-S (Liu et al., 2021) | UperNet@160k | $512^2$ | 81 | 47.6 | 49.5 |
| Vim-T (Zhu et al., 2024) | UperNet@160k | $512^2$ | 13 | 41.0 | - |
| Vim-S (Zhu et al., 2024) | UperNet@160k | $512^2$ | 46 | 44.9 | - |
| LocalVim-T (Huang et al., 2024) | UperNet@160k | $512^2$ | 36 | 43.4 | 44.4 |
| LocalVim-S (Huang et al., 2024) | UperNet@160k | $512^2$ | 58 | 46.4 | 47.5 |
| VMamba-T (Liu et al., 2024) | UperNet@160k | $512^2$ | 62 | 47.9 | 48.8 |
| **GlobalMamba-T (ours)** | UperNet@160k | $512^2$ | 62 | **48.1** | **49.0** |
| VMamba-S (Liu et al., 2024) | UperNet@160k | $512^2$ | 82 | 50.6 | 51.2 |
| **GlobalMamba-S (ours)** | UperNet@160k | $512^2$ | 82 | **50.9** | **51.4** |
| VMamba-B (Liu et al., 2024) | UperNet@160k | $512^2$ | 122 | 51.0 | 51.6 |
| **GlobalMamba-B (ours)** | UperNet@160k | $512^2$ | 122 | **51.2** | **51.7** |

## 4.4 EXPERIMENTAL ANALYSIS

**Causal Order.** The causal modeling sequence from low to high frequency is the prior imposed by our GlobalMamba. To demonstrate the rationality and effectiveness of this sequence, we compare the performance of frequency division methods from high to low frequency and with randomly selected frequency intervals. The specific methods of the three frequency divisions and the performance comparison are shown in Figure 4. We see that randomly selecting the range for frequency division is detrimental to the classification accuracy of the model, and the performance gain from the high-to-low frequency sequence is significantly less than that of the low-frequency prior criterion adopted by GlobalMamba.

**Number of Frequency Segments.** GlobalMamba performs multi-segment frequency division to obtain the corresponding causal sequences. Therefore the number of frequency bands $K$ is a crucial factor, representing the granularity of frequency division and directly determining the length of the causal sequences. To this end, we investigated the impact of different division numbers on

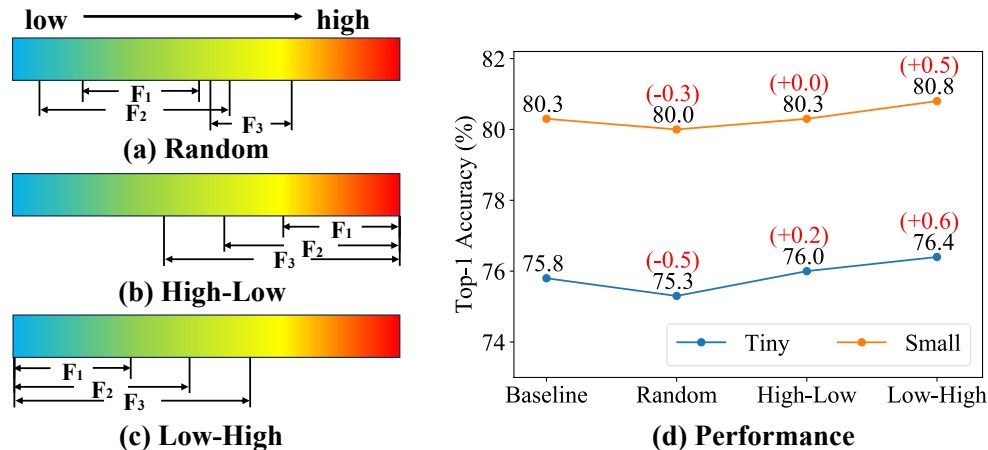

Figure 4: Effect of the causal order: (a) Random division of frequency bands. (b) Dividing the frequency bands in descending order from high to low frequency. (c) Dividing the frequency bands in descending order from low to high frequency. (d) The corresponding classification performances.

Table 5: Effect of the segment number.

| Method | K | Length | Size | Top-1 Acc | Size | Top-1 Acc |
|---|---|---|---|---|---|---|
| Vim | - | 197 | Tiny | 75.8 | Small | 80.3 |
| Vim | - | 393 | Tiny | 75.0 | Small | 79.2 |
| GlobalMamba* | 2 | 246 | Mini | **75.9** | Tiny | **80.5** |
| GlobalMamba* | 3 | 255 | Mini | **76.2** | Tiny | **80.7** |
| GlobalMamba* | 4 | 256 | Mini | **76.4** | Tiny | **80.8** |
| GlobalMamba* | 5 | 257 | Mini | **76.3** | Tiny | **80.9** |
| GlobalMamba* | 6 | 258 | Mini | **76.4** | Tiny | **80.9** |

Table 6: Application to the Causal Transformer.

| Method | Type | Top-1 Acc |
|---|---|---|
| CausalT-S | Plain | 72.2 |
| CausalT-S + GIS | Plain | **73.0** |
| CausalT-S | Pyramid | 75.0 |
| CausalT-S + GIS | Pyramid | **75.5** |

model performance, and also provided a performance comparison of the Vim baseline when directly replicating and augmenting the sequence length in Table 5. Firstly, we verify that directly replicating tokens in Vim fails to bring performance improvement and even reduces the accuracy of the original model. Additionally, we observe that as the value of $K$ increases from 2 to 6, the classification performance rises first and then stabilize. Specially, decent performance is achieved when $K = 4$ for both model sizes and further enlarging $K$ will slightly increase the sequence length but will not result in significant performance gains. Therefore, we set $K$ to 4 in the main experiments.

**Application to Causal Transformer.** In addition to vision mambas, decoder-only transformers also possess the capability for causal modeling of inputs. Therefore, we tested the effectiveness of the proposed global image serialization (GIS) approach on causal transformer by modifying the original self-attention mechanism of DeiT-S and Swin-T to a causal form and applying it to ImageNet classification in Table 6. The consistent performance improvement in both the plain and pyramid types of causal transformer structures demonstrates the flexibility and superiority of our GlobalMamba.

## 5 CONCLUSION

In this paper, we have proposed GlobalMamba as an effective visual backbone for representation learning. We have adopted DCT to perform the corresponding frequency band arrangement in the frequency domain, constructing a series of causal image sequences ranging from low to high frequency. We have further ensured that the token sequence associated with the low-frequency components is capable of extracting global information within the image, thereby significantly enhancing the global comprehension of the visual data. We have validated the effectiveness of GlobalMamba on diverse vision tasks and conducted in-depth ablation studies for detailed analysis and comparison.

**Limitations.** Due to the lower downsampling rate corresponding to the high-frequency components, there still exists a portion of flattening operations. Future work will focus on completely avoiding such simple flattening to obtain a more robust causal sequence.

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
