# OpenReview forum: "GlobalMamba: Global Image Serialization for Vision Mamba"
_ICLR.cc/2025/Conference — ICLR 2025 Conference Withdrawn Submission_

### Official Review · Reviewer_zaAz · 2024-10-30

**Soundness:** 2
**Presentation:** 3
**Contribution:** 2
**Rating:** 5
**Confidence:** 4

**Summary:**

The work builds on Mamba state space models (SSMs) in vision tasks. State space models have been proposed as sequence models that can improve the limitations of quadratic complexity in self-attention by leveraging a framework that allows for parallelism in training, with sequential predictions in autoregressive tasks. In the field of vision, research on SSMs for vision has focused on remediating a central limitation of sequential modelling of spatial data; optimal ways for aligning data in a flattened two-way message passing setup.

The authors bring about an important point; given that spatial data have less of an inherent ordering than general sequence modelling tasks, it is not clear how a global receptive field can be achieved as handily as in vision transformers (ViTs). For transformers, a bidirectional approach simplifies to a global view where all tokens interact in parallel per self-attention block. Contrarily, bidirectionality in more classic sequence models still implies two-way message passing in a flattened sequence. Hence, work on SSMs for vision have rediscovered classic scan orders over patches to determine an appropriate extension to modelling spatial data as sequences. Currently, SSMs use more layers with lower throughput to achieve typically slightly worse results than ViTs.

The authors go on to propose an approach to provide global context for SSMs in vision by introducing tokenization by a pyramidal structure of low-pass filtered tokens from the top down. While the paper touches on a central modelling component in SSMs for vision, the authors end up modelling the pyramidal tokens at each level with classic raster scan ordering, referred to as “causal tokens”. The resulting method ends up being better, however, it is very unclear if this is due to the arguments in the paper. Instead, the gains could feasibly come from having many more tokens in the sequence, which generally improves performance, but increases complexity. **In other words, showing that the gains come from the ordering of low-pass filtered images rather than the feature pyramid with increased sequence length becomes imperative to demonstrate**, given the central argument provided by the authors.

**In summary**, it seems that the tokenization method has benefits to other Mamba-based approaches in vision tasks. However, in this reviewer's opinion, the current draft does not adequately delineate the source of these performance gains. As it stands, it seems possible that gains can be attributed to more tokens, rather than a causal ordering of frequency bands in the image.

**Strengths:**

S1: The method provides quite significantly stronger results compared to other Mamba-based SSNs, with results for both image level and dense prediction tasks and models for various model capacities.

S2: Tokenization is an important and often overlooked component in both transformers and SSMs for vision tasks, and the approach has merit as a pyramidal tokenizer for both modelling paradigms (SSNs and ViTs). This reviewer appreciates the author's contribution to this field.

S3: The approach utilises tried and true techniques in image processing, even if the manuscript in its current form does not reflect the extensive history in the background of the proposed method.

S4: While the approach adds more tokens to the sequence, this does lean into the proposed benefits of longer sequences that SSNs can provide.

**Weaknesses:**

W1: In contrast to how the method is motivated, the method does not subdivide the image into frequency bands, but instead employs a low-pass image pyramid. This has the effect of increasing the overall number of tokens in the sequence compared to other baselines, which is not mentioned in the paper. In the end, the tokenization method mimics classic Gaussian pyramids (which are not cited or referenced in the work).

W2: While the authors argue that the approach provides  “causal tokens”, the grounding in causality is somewhat opaque. While it could be argued that the tokenization approach is more “global” through the pyramidal structure, it still ends up relying on a scan order in each level, which is arbitrary.

W3: It is not clear that frequency subsampling in the DCT domain plays a significant role in the proposed pyramid structure, and thus the arguments on causality in the frequency domain are substantially weaker. Instead, the increase in performance seems more likely to stem from including many more tokens in the sequence.

W4: It must be noted that there is a degree of ahistoricity in the presentation of the proposed method. The paper makes little effort to place their work in the context of the classical methods they end up employing.

**More minor weaknesses**:

W5: While the overall FLOPs of the method is provided, there is no mention of throughput, which could be a central factor for modellers looking to employ the proposed method. Lower FLOPs do not always translate to faster inference, particularly in the case of Mamba models.

W6: The paper spends an inordinate amount of space outlining the fundamentals of the DCT, which should be clear to any reader with any tangentially relevant background in image processing. This could be moved to the appendix or supplementary material.

W7: The approach relies on somewhat older ViT baselines. While it is appropriate to compare against models that are exclusively trained on ImageNet1k, there are stronger supervised models which would provide a more fair comparison to the proposed model available.

W8: The paper has a few typos and missing references to figures. This is a very minor point and does not impact the review or score.

**Questions:**

Q1: A central issue with the current paper is linking the approach to the argument of "causal tokens". Did you ablate the effect of the low-pass DCT filtering process during your experiments? In other words, can you show that this is the source of the gains in the model? Given the pyramidal structure, this reviewer struggles to find any particular reason why simply downscaling patches in the pyramid without applying a low-pass filter should fail to achieve exactly the same results.

Q2: Given the extensive history of the techniques employed by the authors, are there any reason to omit references to these, or are the authors unaware of the overlap with classical image modelling approaches? In particular, Gaussian pyramids (Adelson et al. 1984) have a long history that is easily contextualised. The approach of sequentially scanning frequencies diagonally starting from the top right corner seems to draw inspiration from classic DCT compression regimes, such as JPEG (Wallace, 1992).

Q3: Why is DEiT chosen as an appropriate baseline for comparison instead of the more recent DEiTv3? Notably, DEiT uses distillation from a convolutional model, while DEiTv3 is a pure transformer architecture.

Q4: Can you provide examples of throughput (images / sec) for your models compared to baselines?

---

### Official Review · Reviewer_DWD1 · 2024-10-31

**Soundness:** 2
**Presentation:** 3
**Contribution:** 2
**Rating:** 5
**Confidence:** 4

**Summary:**

This manuscript highlights a limitation in current vision mambas, which utilize tokenization and a sequential flattening process. The process overlooks the inherent 2D structural correlations present in images, making it challenging to extract global information effectively. To address these limitations, the proposed GlobalMamba method strategically organizes pixels according to their corresponding Discrete Cosine Transform (DCT) frequencies. Subsequently, it transforms each set of pixels with the same frequency back into the spatial domain, resulting in a longer image series for the subsequent tokenization in a mamba baseline. The experiments tackling tasks of image classification, object detection, and semantic segmentation are conducted on ImageNet-1K, COCO, and ADE20K datasets.

**Strengths:**

+ The manuscript is clear and well-structured, making it easy for readers to understand its content.
+ The proposed frequency-based image serialization method is straightforward and can be readily reimplemented.

**Weaknesses:**

- The exploration of extracting information from the frequency domain in relation to mamba schemes is one topic of recent research efforts [A, B]. This paper introduces an alternative method for utilizing frequency information. However, the experimental findings suggest that the performance improvements achieved through the proposed GlobalMamba are relatively minimal. Notably, the increase in percentage in FLOPS surpasses the enhancement in accuracy, raising questions about the actual benefits of the proposed frequency-based image serialization for downstream tasks. For comparison, the long sequence version of Vim-T improves Vim-T on Top-1 ACC from 76.1 to 78.3, yet the proposed GlobalMamba improves the reproduced Vim-T from 75.8 to 76.4.

- The findings presented in Table 5 indicate that the use of replicated and augmented token sequences negatively impacts the model's performance. This raises questions about the effectiveness of the proposed image serialization approach, which appears to duplicate the image information. It is intriguing to consider the underlying reasons for the varied results observed. It would be better to have a more detailed analysis of why GlobalMamba's performance is better than simple replication while both approaches are increasing sequence length.

Related papers:
 - [A] Yi Xiao, Qiangqiang Yuan, Kui Jiang, Yuzeng Chen, Qiang Zhang, Chia-Wen Lin: Frequency-Assisted Mamba for Remote Sensing Image Super-Resolution. TIP (2024)
- [B] Zou Zhen, Yu Hu, Zhao Feng: FreqMamba: Viewing Mamba from a Frequency Perspective for Image Deraining. ACMMM (2024)

**Questions:**

The implementation of the proposed GlobalMamba is relatively straightforward. However, a significant concern arises regarding its minor model performance, which does not adequately support the benefits of the proposed frequency-based image serialization for Mamba. When considering the associated increase in computational costs, the slight improvement in performance appears to be an insufficient contribution overall. If the benefits of GlobalMamba become more pronounced for certain types of images or tasks, it would be great to discuss them further in the manuscript.

---

### Official Review · Reviewer_TjQW · 2024-11-04

**Soundness:** 2
**Presentation:** 3
**Contribution:** 2
**Rating:** 5
**Confidence:** 3

**Summary:**

The paper introduces GlobalMamba, a visual model designed to improve image representation by arranging image tokens in a frequency-based, causal sequence. The model leverages the Discrete Cosine Transform (DCT) to decompose images into different frequency bands, which are transformed back to the spatial domain to form image sequences progressing from low to high frequency. This sequence is then processed by a Mamba-based model to enhance feature extraction by focusing first on global, low-frequency information before finer details.

**Strengths:**

The idea of frequency-based tokenization is interesting. It enables the model to retain more global context, particularly in the low-frequency bands, aligning with the frequency principle and mimicking human visual processing.

**Weaknesses:**

1. Marginal Performance Improvement. The observed accuracy gains are relatively minor, often below 1%, despite the increase in model complexity and especially the increase of FLOPs.

2. Higher Computational Cost. While downsampling can mitigate some of the overhead, the sequence length and computational demand increase, raising questions about efficiency versus benefit for limited gains.

3. Complexity is also Higher. The frequency-based segmentation and DCT/IDCT transformations add structural complexity, which complicates the deployment and integration compared to simpler tokenization methods.

**Questions:**

The reviewers' concern is mainly about the increased computation costs and complexity. While FLOPs are kept in check, the author could provide a more detailed analysis of computational costs relative to baseline improvements, as the gains in accuracy may not justify the added computational complexity.

---

### Official Review · Reviewer_CmEA · 2024-11-04

**Soundness:** 2
**Presentation:** 3
**Contribution:** 3
**Rating:** 5
**Confidence:** 4

**Summary:**

Most of existing Vision Mamba frameworks employ patch-based image tokenization and then flatten them into 1D sequences for causal processing, which is not effective in processing 2D structural correlations of images. With a frequency-based image serialization method, this paper proposes a new vision mamba model, GlobalMamba, which can better exploit 2D structural correlations of image token sequences. Experiments on image classification, object detection and semantic segmentation are conducted to demonstrate the effectiveness of the proposed method.

**Strengths:**

- Overall, the paper is well-written and easy to follow.
- The motivation is clear. The proposed approach is reasonable.

**Weaknesses:**

-	The method requires more FLOPs than V-Mamba. The performance improvement over V-Mamba is somewhat limited. On most benchmarks, the improvement over V-Mamba is only 0.2%-0.3%.
-	In Table 2, The result of GlobalMamba-M (Mini) is marked in bold. However, GlobalMamba-M (Mini) seems to have lower classification accuracy than PlainMamba-L1, EffVMamba-T and EffVMamba-S. Especially, EffVMamba-T is also more efficient in Params and FLOPs. Why EffVMamba is more efficient in the mini size?

**Questions:**

Please see above.

---

### Official Review · Reviewer_B8XY · 2024-11-05

**Soundness:** 3
**Presentation:** 3
**Contribution:** 3
**Rating:** 5
**Confidence:** 5

**Summary:**

This paper tries to address the inherent limitations of current Vision Mamba models, which achieve linear complexity by processing image tokens sequentially but rely on patch-based tokenization that flattens 2D images into 1D sequences, thereby neglecting essential 2D structural correlations and hindering the extraction of global information. To overcome these challenges, the authors introduce a novel global image serialization method that utilizes the Discrete Cosine Transform (DCT) to convert images from the spatial to the frequency domain, organizing pixels based on their frequency ranges. By transforming each frequency band back to the spatial domain, they generate a series of images that preserve global information for effective tokenization. Building on this approach, the proposed GlobalMamba model employs a causal input format that better captures the causal relationships within image sequences. Extensive experiments demonstrate that GlobalMamba significantly outperforms existing Vision Mamba models and Vision Transformers in key vision tasks, including image classification on ImageNet-1K, object detection on COCO, and semantic segmentation on ADE20K. This work presents a meaningful advancement in leveraging global image information within Vision Mamba architectures, enhancing their applicability and performance across diverse high-resolution and complex vision tasks.

**Strengths:**

1. GlobalMamba orders image tokens by frequency, enabling the model to first capture basic structures like contours and then add finer details, mimicking the way "humans process visual information".

**Weaknesses:**

1. Lack of reporting inference time. Since the feature maps in stage 1 of GlobalMamba-T/S/B have high resolution, the inference time would be the concern. Can you report the inference time?
2. Limited performance gain. Compared with Vmamba, GlobalMamba brings limited performance gain in classification, segmentation, detection.

I am happy to raise my score if the authors can address my concerns.

**Questions:**

What if GlobalMamba drop low/high frequency information? Will this hurt the performance?

---

### Note · Authors · 2024-11-15

I have read and agree with the venue's withdrawal policy on behalf of myself and my co-authors.